# Training Neural Machines with Partial Traces

## Abstract

We present a novel approach for training neural machines which incorporates additional supervision on the machine's interpretable components (e.g., neural memory). To cleanly capture the kind of neural machines to which our method applies, we introduce the concept of a differential neural computational machine ($\partial$NCM) and show that several existing architectures (e.g., NTMs, NRAMs) can be instantiated as a $\partial$NCM and can thus benefit from *any* amount of additional supervision over their interpretable components. Based on our method, we performed a detailed experimental evaluation with NTM and NRAM machines, showing the approach leads to significantly better convergence and generalization capabilities of the learning phase than standard training using only input-output examples.

## 1 Introduction

Recently, there has been substantial interest in neural abstract machines that can induce programs from examples Feser et al. (2015; 2016); Frankle et al. (2016); Gaunt et al. (2017); Graves et al. (2014; 2016); Kaiser & Sutskever (2016); Kurach et al. (2016); Reed & de Freitas (2016); Vinyals et al. (2015); Zaremba & Sutskever (2015); Zhang et al. (2015). While significant progress has been made towards learning interesting algorithms Graves et al. (2016), ensuring the training of these machines converges to the desired solution can be very challenging. Interestingly however, even though these machines differ architecturally, they tend to rely on components (e.g., neural memory) that are more interpretable than a typical neural network (e.g., an LSTM). A key question then is:

*Can we somehow provide additional amounts of supervision for these interpretable components during training so to bias the learning towards the desired solution?*

In this work we investigate this question in depth. We refer to the type of supervision mentioned above as partial trace supervision, capturing the intuition that more detailed information, beyond input-output examples, is provided during learning. To study the question systematically, we introduce the notion of a differential neural computational machine ($\partial$NCM), a formalism which allows for clean characterization of the neural abstract machines that fall inside our class and that can benefit from *any* amount of partial trace information. We show that common architectures such as Neural Turing Machines (NTMs) and Neural Random Access Machines (NRAMs) can be phrased as $\partial$NCMs, useful also because these architectures form the basis for many recent extensions, e.g., Graves et al. (2016); Grefenstette et al. (2015); Kaiser & Sutskever (2016). We also explain why other machines such as the Neural Program Interpreter (NPI) Reed & de Freitas (2016) or its recent extensions (e.g., the Neural Program Lattice Li et al. (2017)) cannot be instantiated as an $\partial$NCM and are thus restricted to require large (and potentially prohibitive) amounts of supervision. We believe the $\partial$NCM abstraction is a useful step in better understanding how different neural abstract machines compare when it comes to additional supervision. We then present $\partial$NCM loss functions which abstractly capture the concept of partial trace information and show how to instantiate these for both the NTM and the NRAM. We also performed an extensive evaluation for how partial trace information affects training in both architectures. Overall, our experimental results indicate that the additional supervision can substantially improve convergence while leading to better generalization and interpretability.

To provide an intuition for the problem we study in this work, consider the simple task of training an NTM to flip the third bit in a bit stream (called `Flip3rd`) – such bitstream tasks have been extensively studied in the area of program synthesis (e.g., Jha et al. (2010); Raychev et al. (2016)). An example input-output pair for this task could be $[0, 1, 0, 0] \rightarrow [0, 1, 1, 0]$. Given a set of such

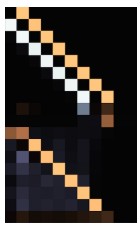 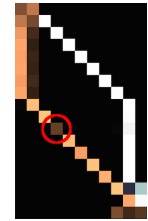 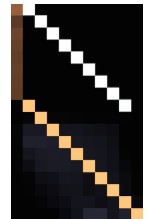

(a) Overfitting & No Traces      (b) Generalizing & No Traces      (c) Generalizing & Traces

Figure 1: Traces of locations accessed by read/write heads for the `Flip3rd` task in three different training setups. The y-axis represents time (descending); x-axis represents head locations. First two NTMs are trained without partial trace information. White represents the distribution of the write head; orange the distribution of the read head; (b) and (c) generalize and are more interpretable than (a); (c) was trained using partial trace information and is more interpretable than (b).

examples, our goal is to train an NTM that solves this task. An example NTM that generalizes well and is understandable is shown in Figure 1c. Here, the y-axis is time (descending), the x-axis is the accessed memory location, the white squares represent the write head of the NTM, and the orange squares represent the read head. As we can see, the model writes the input sequence to the tape and then reads from the tape in the same order. However, in the absence of richer supervision, the NTM (and other neural architectures) can easily overfit to the training set – an example of an overfitting NTM is shown in Figure 1a. Here, the traces are chaotic and difficult to interpret. Further, even if the NTM generalizes, it can do so with erratic reads and writes, an example of which is shown in Figure 1b. Here, the NTM learns to read from the third bit (circled) with a smaller weight than from other locations, and also reads and writes erratically near the end of the sequence. This model is less interpretable than the one in Figure 1c because it is unclear how the model knows which the third bit actually is, or why a different read weight would help flip that bit.

In this work we will develop principled ways for guiding the training of a neural abstract machine towards the behavior shown in Figure 1c. For instance, for `Flip3rd`, providing partial trace information on the NTM's read heads for 10% of the input-output examples is sufficient to bias the learning towards the NTM shown in Figure 1c 100% of the time.

## 2 NEURAL COMPUTATIONAL MACHINES

To capture the essence of our method and illustrate its applicability, we now define the abstract notion of a *neural computational machine* (NCM). NCMs mimic classic computational machines with a controller and a memory, and generalize multiple existing architectures. Our approach for supervision with partial trace information applies to all neural architectures expressible as NCMs. A useful feature of the NCM abstraction is that it clearly delineates end-to-end differentiable architectures (Graves et al. (2014)'s NTM, Kurach et al. (2016)'s NRAM), which can train with little to no trace supervision, from architectures that are not end-to-end differentiable (Reed & de Freitas (2016)'s NPI) and hence require a certain minimum amount of trace information. In the follow-up section, we show how to phrase two existing neural architectures (NTMs and NRAMs) as an NCM.

An *NCM* is a triple of functions: a processor, a controller, and a loss:

**Processor** The processor $\pi : W \times C \times M \to B \times M$ performs a pre-defined set of commands $C$, which might involve manipulating memories in $M$. The commands may produce additional feedback in $B$. Also, the processor's operation may depend on parameters in $W$.

**Controller** The controller $\kappa : W \times B \times Q \times I \to C \times Q \times O$ decides which operations the machine performs at each step. It receives external inputs from $I$ and returns external outputs in $O$. It can also receive feedback from the processor and command it to do certain operations (e.g., memory read). The decisions the controller takes may depend on its internal state (from $Q$). The controller can also depend on parameters in $W$. For instance, if the controller is a neural network, then the network's weights will range over $W$.

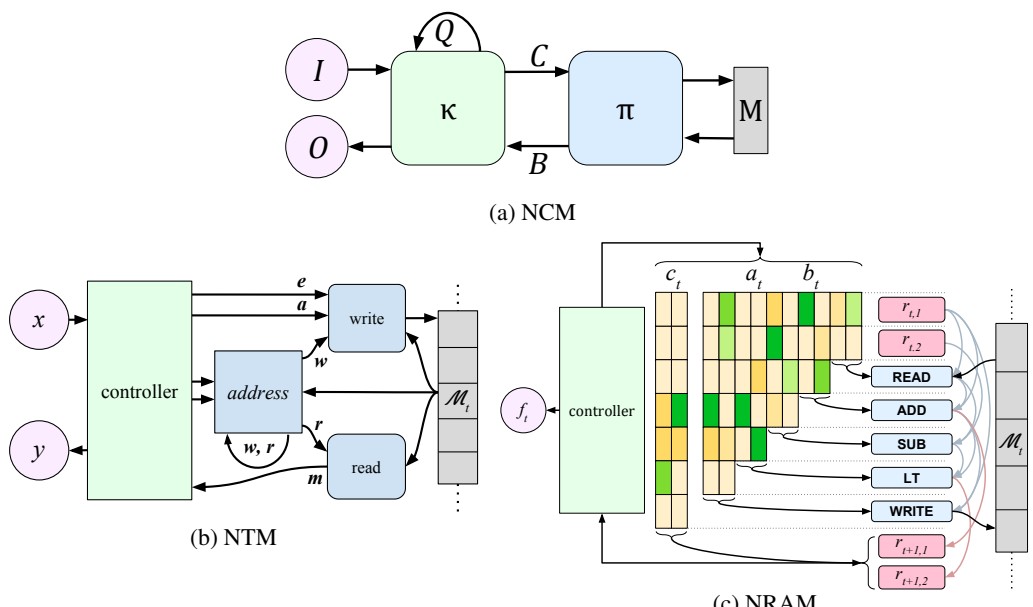

Figure 2: (a) is depiction of the generic NCM structure (b) is a high-level overview of an NTM and (c) is a high level overview of the NRAM architecture. The controller outputs a circuit, which in this case contains the modules READ, ADD, SUB, LT, and WRITE. The controller encodes the two inputs to the modules as probability vectors, $a, b$ and $c$, over the possible choices. The most likely choice is shown in green. The only input to the controller are the registers $r_1$ and $r_2$.

**Loss Function** The loss function $L_e : Trace \times E \to \mathbb{R}$ indicates how close a trace $\tau \in Trace$ of an execution of the machine (defined below) is to a behavior from a set $E$. The loss function provides a criterion for training a machine to follow a prescribed set of behaviors, and hence we impose certain differentiability conditions. We require that the loss surface is continuous and piecewise differentiable with respect to the weights $w \in W$ for all examples $e$ and inputs $x$ with traces $\tau(w, x)$:

$$l(w; x, e) = L_e(\tau(w, x), e) \tag{1}$$

**Execution** The execution of the machine begins with an input sequence $x = \{x_t\}_1^n$ and initial values of the controller state $q_0$, memory $m_0$, and processor feedback $b_0$. At each time step $t = 1 \ldots n$, controller and processor take turns executing according to the following equations:

$$
\begin{aligned}
(c_t, q_t, y_t) &= \kappa(w, b_{t-1}, q_{t-1}, x_t) \\
(b_t, m_t) &= \pi(w, c_t, m_{t-1})
\end{aligned}
\tag{2}
$$

A *trace* $\tau(w, x, b_0, m_0, q_0) = \{(c_t, b_t, q_t, y_t, m_t)\}_1^n$ records these quantities' values at each time step. We will occasionally write $\tau_C, \tau_B, \ldots$ for the trace projected onto one of its components $c, b, \ldots$.

$\partial$**NCMs** Note that the differentiability conditions that we impose on the loss do not imply that any of the NCM functions $\pi$, $\kappa$ and $L_e$ are continuous or differentiable. They indeed can be highly discontinuous as in NCMs like Weston et al. (2014)'s memory networks with a hard attention mechanism, or as in Reed & de Freitas (2016)'s neural programmer-interpreters. In order to fix these discontinuities and recover a differentiable loss surface, these architectures train with strong supervision only: the training examples $e \in E$ must provide a value for every traced quantity that comes from a discontinuous parameter.

In contrast, what we call *differentiable neural computational machines* ($\partial$NCM), have $\kappa$, $\pi$ and $L_e$ continuous and piecewise differentiable. In this case, the loss surface is differentiable with respect to every parameter. Thus, there is no need to specify corresponding values in the examples, and so we can train with as much trace information as available.

## 3 NTMs and NRAMs as NCMs

We now show how NTMs and NRAMs can be instantiated as $\partial$NCMs.

**NTM as $\partial$NCM**  An Neural Turing Machine (NTM) Graves et al. (2014) (Figure 2b) has access to a memory $\mathcal{M} \in \mathbb{R}^{c \times n}$ of $c$ cells of $n$ real numbers each. We suppose the machine has one read head and one write head, whose addresses are, respectively, the probability vectors $r, w \in [0, 1]^{\{1 \ldots c\}}$. At every time step, the read head computes the expected value $m \in \mathbb{R}^n$ of a random cell at index $i \sim r$. This value together with the current input are fed into a controller neural network, which then decides on several commands. It decides what fraction $e \in \mathbb{R}^n$ to erase and how much $a \in \mathbb{R}^n$ to add to the cells underneath the write head. The write head stores the tape expected after a random modification at index $i \sim w$. Then the controller indicates the head movement with two probability vectors $\Delta r, \Delta w \in [0, 1]^{\{-1, 0, +1\}}$ which are convolved with the respective head addresses (the actual addressing mechanism is more involved, but we omit it for brevity) Finally, the controller produces the current output value. In terms of NCMs, the NTM's variables fall into the following classes:

| I/O | State | Communication |
|---|---|---|
| $x \in I$ | $q \in Q$ | $(e, a, \Delta r, \Delta w) \in C$ |
| $y \in O$ | $(r, w, \mathcal{M}) \in M$ | $m \in B$ |

Each of these variables change over time according to certain equations (see Appendix A for details). The processor $\pi$ and the controller $\kappa$ functions for each time step satisfy:

$$
\begin{aligned}
((e_t, a_t, \Delta r_t, \Delta w_t),\ q_t,\ y_t) &= \kappa(w, m_t, q_{t-1}, x_t) \\
(m_{t+1},\ (r_t, w_t, \mathcal{M}_t)) &= \pi((e_t, a_t, \Delta r_t, \Delta w_t),\ (r_{t-1}, w_{t-1}, \mathcal{M}_{t-1})).
\end{aligned}
\tag{3}
$$

The standard loss function $L_e$ for the NTM simply includes a term, such as cross-entropy or $L_2$ distance, for the machine output at every time step. Each of these compare the machine output to the respective values contained in the examples $e \in E$.

**NRAM as $\partial$NCM**  A Neural Random Access Machine (NRAM) Kurach et al. (2016) is a neural machine designed for ease of pointer (de-)referencing. An NRAM has a variable sized memory $\mathcal{M} \in \mathbb{R}^{c \times c}$ whose size varies between runs. It also has access to a register file $r \in \mathbb{R}^{n \times c}$ with a constant number $n$ of registers. Both the memory and the registers store probability vectors over $\{1 \ldots c\}$. The controller receives no external inputs, but at each time step reads the probability that a register assigns to 0. It also produces no external output, except a probability $f \in [0, 1]$ for termination at the current time step. The output of the run is considered to be the final memory state.

Unlike the NTM, computation in the NRAM is performed by a fixed sequence of modules. Each module implements a simple integer operation/memory manipulation lifted to probability vectors. For example, addition lifts to convolution, while memory access is like that of the NTM. At every time step the controller organizes the sequence of modules into a circuit, which is then executed. The circuit is encoded by a pair of probability distributions per module, as shown in Figure 2c. These distributions specify respectively which previous modules or registers will provide a given module first/second arguments. The distributions are stacked in the matrices $a$ and $b$. A similar matrix $c$ is responsible for specifying what values should be written to the registers at the end of the time step. The NCM instantiation of an NRAM is the following:

| I/O | State | Communication |
|---|---|---|
| $\{1\} = I$ | $q_t \in Q$ | $(a_t, b_t, c_t) \in C$ |
| $f_t \in O$ | $(r_t, \mathcal{M}_t) \in M$ | $r_{t, -, 0} \in B$ |

The equations that determine these quantities can be found in Appendix B. The processor function $\pi$ and the controller function $\kappa$ expressed in terms of these quantities are:

$$
\begin{aligned}
((a_t, b_t, c_t), h_t, f_t) &= \kappa(w, r_{(t-1), -, 0}, h_{t-1}, 1) \\
(r_{t, -, 0}, (r_t, \mathcal{M}_t)) &= \pi((a_t, b_t, c_t),\ (r_{t-1}, \mathcal{M}_{t-1})).
\end{aligned}
\tag{4}
$$

The loss of the NRAM is more complex than the NTM loss: it is an expectation with respect to the probability distribution $p$ of termination time, as determined by the termination probabilities $f_t$ (see Appendix B). For every $t = 1 \ldots k$, the loss considers the negative log likelihood that the $i$-th memory cell at that time step equals the value $e_i$ provided in the example, independently for each $i$:

$$L_e(\tau, e) = -\sum_{t<k} p_t \sum_{i<c} \log(\mathcal{M}_{t,i,e_i}). \tag{5}$$

## 4 SUBTRACE SUPERVISION OF NCMS

Incorporating supervision during NCM training can be helpful with: (i) *convergence*: additional bias may steer the minimization of the NCM's loss function $L_e$, as much as possible, away from local minima that do not correspond to good solutions, (ii) *interpretability*: the bias can also be useful in guiding the NCM towards learning a model that is more intuitive/explainable to a user (especially if the user already has an intuition on what it is that parts of the model should do), and (iii) *generalization*: the bias can steer the NCM towards solutions which minimize not just the loss on example of difficulties it has seen, but on significantly more difficult examples.

The way we provide additional supervision to NCMs is, by encoding, for example, specific commands issued to the processor, into extra loss terms. Let us illustrate how we can bias the learning with an NTM. Consider the task of copying the first half of an input sequence $\{x_t\}_1^{2l}$ into the second half of the machine's output $\{y_t\}_1^{2l}$, where the last input $x_l$ from the first half is a special value indicating that the first half ended. Starting with both heads at position 1, the most direct solution is to consecutively store the input to the tape during the first half of the execution, and then recall the stored values during the second half. In such a solution, we expect the head positions to be:

$$w_t = p(t) = \begin{cases} \text{one-hot}(t) & \text{if } t = 1 \ldots l \\ \text{one-hot}(l) & \text{if } t \geq l+1 \end{cases} \qquad r_t = q(t) = \begin{cases} \text{one-hot}(1) & \text{if } t = 1 \ldots l \\ \text{one-hot}(t-l) & \text{if } t \geq l+1 \end{cases} \tag{6}$$

To incorporate this information into the training, we add loss terms that measure the cross-entropy ($H$) between $p(t)$ and $w_t$ as well as between $q(t)$ and $r_t$. Importantly, we need not add terms for every time-step, but instead we can consider only the corner cases where heads change direction:

$$\sum_{t=1,l+1,2l} H(p(t), w_t) + H(q(t), r_t).$$

### 4.1 GENERIC SUBTRACE LOSS FOR NCMS

We now describe the general shape of the extra loss terms for arbitrary NCMs. Since, typically, we can interpret only the memory and the processor in terms of well-understood operations, we will consider loss terms only for the memory state and the communication flow between the controller and the processor. We leave the controller's hidden state unconstrained – this also permits us to use the same training procedure with different controllers.

The generic loss is expressed with four loss functions for the different components of an NCM trace:

$$\begin{aligned} L_C : C \times E_C &\to \mathbb{R} & L_B : B \times E_B &\to \mathbb{R} \\ L_O : O \times E_O &\to \mathbb{R} & L_M : M \times E_M &\to \mathbb{R} \end{aligned} \tag{7}$$

For each part $\alpha \in \{C, B, O, M\}$, we provide *hints* $(t, v, \mu) \in \sigma_\alpha$ that indicate a time step $t$ at which the hint applies, an example $v \in E_\alpha$ for the relevant component, and a weight $w \in \mathbb{R}$ of the hint. The weight is included to account for hints having a different importance at different time-steps, but also to express our confidence in the hint, e.g., hints coming from noisy sources would get less weight.

A *subtrace* $\sigma$ is a collection of hints used for a particular input-output example $e$. We call it a subtrace because, typically, it contains hints for a proper subset of the states traced by the NCM during execution. The net loss for a given input-output example and subtrace equals the original loss $L_e$ added to the weighted losses for all the hints, scaled by a constant factor $\lambda$:

$$L(\tau, (\sigma, e)) = L_e(\tau, e) + \lambda \frac{\sum_{\alpha \in \{C,B,O,M\}} \sum_{(t,v,\mu) \in \sigma_\alpha} \mu L_\alpha(\tau_{\alpha,t}, v)}{\sum_{\alpha \in \{C,B,O,M\}} \sum_{(t,v,\mu) \in \sigma_\alpha} \mu} \tag{8}$$

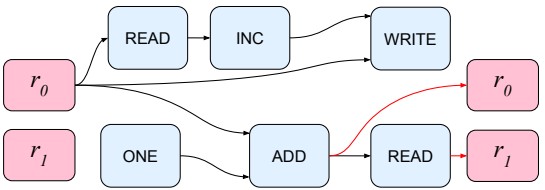

Figure 3: An example circuit for the task of adding one to all memory cells. The arrows for register updates are shown in red. Technically, modules take two arguments, but some ignore an argument, such as INC or READ. For them, we show only the relevant arrows.

## 4.2 SUBTRACES FOR NTM

For NTMs, we allow hints on the output $y$, the addresses $r$ and $w$, and the tape $\mathcal{M}$. We include extra loss terms for the memory state only (all other loss terms are zero):

$$L_M((r_t, w_t, \mathcal{M}_t), (\texttt{wr}, v)) = H(v, w_t)$$
$$L_M((r_t, w_t, \mathcal{M}_t), (\texttt{rd}, v)) = H(v, r_t) \tag{9}$$

Unlike the output and addresses, values on the tape are interpreted according to an encoding internal to the controller (which emerges only during training). Forcing the controller to use a specific encoding for the tape, as we do with NTM output, can have a negative effect on training (in our experiments, training diverged consistently). To remedy this, we do not apply loss to the tape directly but to a decoded version of a cell on the tape. While a decoder might find multiple representations and overfit, we found that it forced just enough consistency to improve the convergence rate. The decoder itself is an auxiliary network $\phi$ trained together with the NTM, which takes a single cell from memory as input. The output of the decoder is compared against the expected value which should be in that cell:

$$L_M((-, -, \mathcal{M}_t), (\texttt{tape}, i, v)) = H(\phi(\mathcal{M}_{t,i}), v). \tag{10}$$

For all subtraces we provide in our experiments with NTMs, the hints have the same unit weight.

## 4.3 SUBTRACES FOR NRAM

For NRAMs, we hint which connections should be present in the circuit the controller constructs at each step, including the ones for register updates. An example circuit is shown in Figure 3. In terms of an NCM, this amounts to providing loss for commands and no loss for anything else. We set the loss to the negative log likelihood of the controller choosing specific connections revealed in the hint:

$$L_C((a_t, b_t, c_t), (\texttt{module}, m, i, j)) = -\log(a_{t,m,i}) - \log(b_{t,m,j})$$
$$L_C((a_t, b_t, c_t), (\texttt{register}, r, i)) = -\log(c_{t,r,i}) \tag{11}$$

In our experiments, we observed that assigning higher weight to hints at earlier timesteps is crucial for convergence of the training process. For a hint at time-step $t$, we use the weight $\mu = (t+1)^{-2}$. A possible reason for why this helps is that the machine's behavior at later time-steps is highly dependent on its behavior at the early time-steps. Thus, the machine cannot reach a later behavior that is right before it fixes its early behavior. Unless the behavior is correct early on, the loss feedback from later time-steps will be mostly noise, masking the feedback from early time-steps.

**Other Architectures** The NCM can be instantiated to architectures as diverse as a common LSTM network or End-To-End Differentiable Memory Networks. Any programming inducing neural network with at least partially interpretable intermediate states for which the dataset contains additional hints could be considered a good candidate for application of this abstraction.

## 5 EXPERIMENTAL EVALUATION

We evaluated our NCM supervision method on the NTM and NRAM architectures. For each of the two architectures we implemented a variety of tasks and experimented with different setups of trace supervision. The main questions that we address are: (i) does trace supervision help convergence, interpretability, and generalization? (ii) how much supervision is needed to train such models? Below, we summarize our findings – further details are provided in the appendix.

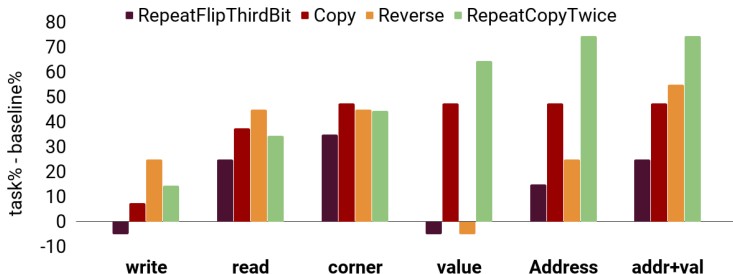

Figure 4: Relative percentage of training instances which generalized out of ten runs per task for the NTM. We provide a subtrace 100% of the time and use $\lambda = 1$. x-axis shows the type of supervision.

| density | | 100 | 100 | 100 | 100 | 50 | 50 | 50 | 50 | 10 | 10 | 10 | 10 | 1 | 1 | 1 | 1 |
|---|---|---|---|---|---|---|---|---|---|---|---|---|---|---|---|---|---|
| $\lambda$ | | 1 | 0.3 | 0.1 | 0.03 | 1 | 0.3 | 0.1 | 0.03 | 1 | 0.3 | 0.1 | 0.03 | 1 | 0.3 | 0.1 | 0.03 |
| baseline | 45 | | | | | | | | | | | | | | | | |
| addr+val | | 30 | 50 | 50 | 50 | 50 | 50 | 70 | 80 | 60 | 80 | 40 | 40 | 40 | 60 | 40 | 10 |
| address | | 0 | 60 | 40 | 40 | 20 | 70 | 80 | 90 | 90 | 70 | 60 | 50 | 50 | 50 | 60 | 60 |
| value | | 60 | 50 | 40 | 60 | 80 | 70 | 40 | 10 | 50 | 10 | 40 | 70 | 50 | 30 | 40 | 60 |
| read | | 40 | 60 | 70 | 40 | 30 | 80 | 90 | 90 | 100 | 70 | 80 | 50 | 30 | 50 | 60 | 30 |
| write | | 0 | 30 | 50 | 20 | 30 | 40 | 60 | 20 | 40 | 40 | 40 | 40 | 20 | 50 | 70 | 50 |
| corner | | 50 | 70 | 80 | 80 | 40 | 40 | 90 | 70 | 80 | 70 | 60 | 40 | 50 | 60 | 60 | 40 |

Figure 5: The number of initial runs which generalized for `Flip3rd`. The first dimension listed in the rows controls the execution details revealed in a subtrace, while the second dimension (the *density* column) controls the proportion of examples that receive extra subtrace supervision.

**Subtraces For NTMs** We measured how often we successfully trained an NTM that achieves strong generalization. We consider a model to generalize if relative to the training size limit $n$, it achieves perfect accuracy on all of tests of size $\leq 1.5n$, and perfect accuracy on 90% of the tests of size $\leq 2n$. Figure 4 reports the average improvement compared to a baseline using only I/O examples. We ran experiments with four different tasks and various types of hints (cf. Appendices C, E). Some of the hint types are: *read* and *write* specify respective head addresses for all time steps; *address* combines the previous two; *corner* reveals the head addresses, but only when the heads change direction; *value* gives value for a single cell. Except for three cases, trace supervision helped improve generalization. Here, `RepeatFlip3d` is most challenging, with baseline generalizing only 5% of the time (cf. Appendix I). Here we have the largest improvement with extra supervision: corner type of hints achieve eight-fold increase in success rate, reaching 40%. Another task with an even larger ratio is `RepeatCopyTwice` (cf. Appendix), where success increases from 15.5% to 100%.

In addition to this experiment, we performed an extensive evaluation of different setups, varying the global $\lambda$ parameter of the loss Eq. 8, and providing hints for just a fraction of the examples. The full results are in Appendix I; here we provide those for `RepeatFlip3d` in Table 5. The table reveals that the efficacy of our method heavily depends on these two parameters. The best results in this case are for the read/corner type of hints $\frac{1}{2}/\frac{1}{10}$ of the time, with $\lambda \in \{0.1, 1\}$. The best results for other tasks are achieved with different setups. Generally, our conclusion is that training with traces 50% of the time usually improves performance (or does not lower it much) when compared to the best method. This observation raises the interesting question of what the best type and amount of hints are for a given task.

Finally, we observed that in all cases where training with trace supervision converged, it successfully learned the head movements/tape values we had intended. This show that trace supervision can bias the architecture towards more interpretable behaviors. In those cases, the NTM learned consistently sharper head positions/tape values than the baseline, as Figure 6 shows for `Flip3rd`.

**Effect of Subtraces For NRAMs** Ease of generalization for the NRAM is a known issue, with Neelakantan et al. (2015) reporting that `ListK` for example generalizes poorly, even when trained with noise in the gradient, curriculum learning, and an entropy bonus. We observed that when run on an indefinite number of examples with the correct number of timesteps and a correct module

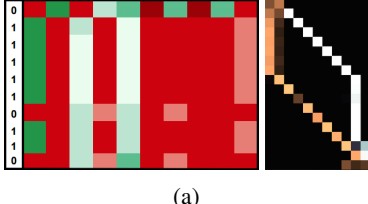 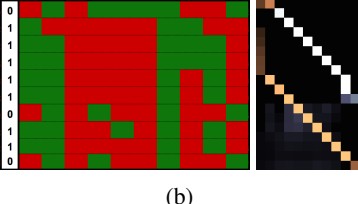

(a)                                          (b)

Figure 6: Execution traces of two NTMs trained on `Flip3rd` until generalization. First is baseline (no trace supervision); second is trained with corner hints. Time flows top to bottom. The first pane from every pair shows the value written to tape; second shows head locations. Figures show that a little extra supervision helps the NTM write sharper 0–1 values and have more focused heads.

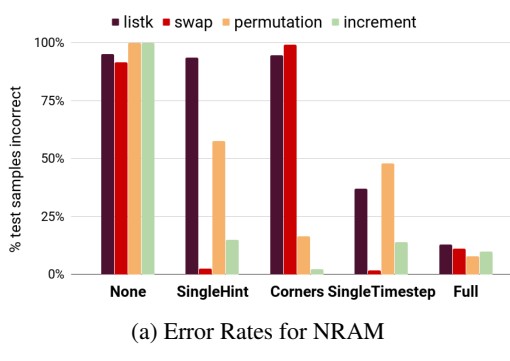

(a) Error Rates for NRAM

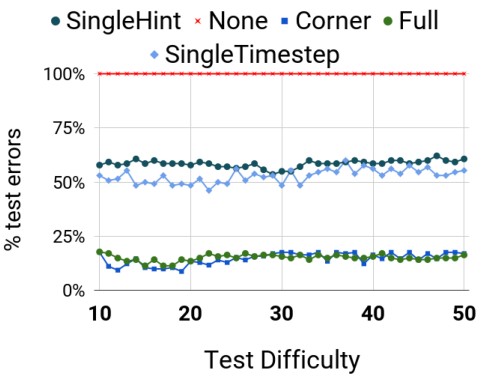

(b) `Permute` with noise for NRAM

Figure 7: (a) average number of errors after training had completed for NRAM. Observe that full training results in a significantly higher percent of generalization after training stopped. (b) shows the distribution of errors to problem length for `Permute` (one character of noise in 10% of samples).

sequence, `Swap` and `Increment` would in fact occasionally generalize perfectly, but did not have the resources to run such indefinite tests with `Permute`, `ListK`, and `Merge`.

Figure 7a demonstrates that when training had finished, either because it had ended early or had reached 5000 training examples (our upper bound), generalization would in fact be on average significantly better than the baseline the more hints that were used for all tasks. Here, number of hints used seemed to be a sufficient predictor for the quality of the trained model.

**Robustness to Noise**  The effect of increasing supervision on the quality of the trained model was so strong that not even noise in the input was able to significantly hinder generalization. In Figure 7b, we corrupted a single character in the output examples for the `Permute` problem in 10% of the examples. We found that without any extra hints, no convergence was seen after training was complete, whereas with just corner subtraces, the generalization was nearly optimal.

Furthermore, we found that noise in the trace does not seriously harm performance. We corrupted a single hint for 20% of the traces of the `Increment` task using otherwise full supervision, as can be seen in the `NoisyFull` line of Figure 14.

## 6  CONCLUSION

We presented a method for incorporating (any amount of) additional supervision into the training of neural abstract machines. The basic idea was to provide this supervision (called partial trace information) over the interpretable components of the machine and to thus more effectively guide the learning towards the desired solution. We introduced the $\partial$NCM architecture in order to precisely capture the neural abstract machines to which our method applies. We showed how to formulate

partial trace information as abstract loss functions, how to instantiate common neural architectures such as NTMs and NRAMs as $\partial$NCMs and concretize the $\partial$NCM loss functions. Our experimental results indicate that partial trace information is effective in biasing the learning of both NTM's and NRAM's towards better converge, generalization and interpretability of the resulting models.

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

## A  NTM Equations

The controller for the NTM consists of the networks $\varphi$, $\psi_y$, $\psi_e$, $\psi_a$, $\chi_r$, $\chi_w$, which operate on the variables:

$$
\begin{array}{llll}
x - \text{in} & q - \text{controller state} & r - \text{read address} & \Delta r - \text{change in } r \quad e - \text{erase} \quad \mathcal{M} - \text{tape} \\
y - \text{out} \quad m - \text{read value} & w - \text{write address} & \Delta w - \text{change in } w \quad a - \text{add} \quad\quad\quad (12)
\end{array}
$$

The equations that describe NTM executions are:

$$
\begin{aligned}
\Delta r_t &= \chi_r(q_t) & r_t &= address(\Delta r_t, r_{t-1}, \mathcal{M}_{t-1}) \\
\Delta w_t &= \chi_w(q_t) & w_t &= address(\Delta w_t, w_{t-1}, \mathcal{M}_{t-1}) \\
y_t &= \psi_y(q_t) & m_t &= r_t \mathcal{M}_{t-1} \\
e_t &= \psi_e(q_t) & \mathcal{M}_t &= \mathcal{M}_{t-1} - (w_t \otimes e_t) \odot \mathcal{M}_{t-1} + w_t \otimes a_t \\
a_t &= \psi_a(q_t) & q_t &= \varphi(x_t, q_{t-1}, m_t).
\end{aligned} \quad (13)
$$

## B  NRAM Equations

The controller of the NRAM consists of the networks $\varphi$, $\psi_a$, $\psi_b$, $\psi_c$, $\psi_f$, which operate on the variables:

$$
\begin{array}{llll}
a - \text{lhs circuit} & b - \text{rhs circuit} & c - \text{register inputs} & o - \text{module outputs} \\
r - \text{register state} & \mathcal{M} - \text{memory tape} & h - \text{controller state} & f - \text{stop probability.} \quad (14)
\end{array}
$$

The equations that describe the NRAM execution are:

$$
\begin{aligned}
a_t &= \text{softmax}(\psi_a(q_t)) & A_{t,i} &= (r_1, \ldots, r_R, o_0, \ldots, o_{i-1})^T a_{t,i} & \forall i < M \\
b_t &= \text{softmax}(\psi_b(q_t)) & B_{t,i} &= (r_1, \ldots, r_R, o_0, \ldots, o_{i-1})^T b_{t,i} & \forall i < M \\
c_t &= \text{softmax}(\psi_c(q_t)) & r_{t,i} &= (r_1, \ldots, r_R, o_1, \ldots, o_Q)^T c_{t,i} & \forall i < R \\
f_t &= \psi_f(q_t) & o_{t,i,k} &= \sum_{0 \le a,b < M} A_{t,i,a} B_{t,i,b}[m_i(a,b) = k] & \forall i \notin \{\rho, \omega\}, k < M \\
q_t &= \varphi(q_{t-1}, r_{t,-,0}) & o_{t,\rho} &= \mathcal{M}_t A_{t,\rho} \\
& & \mathcal{M}_t &= (J - A_{t,\omega})J^T \cdot \mathcal{M}_{t-1} + A_{t,\omega} B_{t,\omega}^T & (15) \\
p_t &= f_t \prod_{i<t}(1 - f_i) & p_T &= 1 - \sum_{i<T} p_i & (16)
\end{aligned}
$$

## C    SETUP FOR NTM

For all of our NTM experiments we use a densely connected feed-forward controller. There are two architectural differences from the original NTM Graves et al. (2014) that helped our baseline performance: (1) the feed-forward controller, the erase and the add gates use `tanh` activation; (2) the output layer uses `softmax`. In the original architecture these are all logistic sigmoids. For the newly introduced tape decoder (active only during training) we used two alternative implementations: a `tanh-softmax` network, and a single affine transformation. We tested the NTM's learning ability on five different tasks for sequence manipulation, two of which have not been previously investigated in this domain. These tasks can be found in Appendix E.

We performed experiments using several combination of losses as summarized in Appendix F. The observed training performance per task is shown in Appendix I, with rows corresponding to the different loss setups. The *corner* setup differs from the *address* setup in that the example subtraces were defined only for a few important corner cases. For example in `RepeatCopyTwice`, the write head was provided once at the beginning of the input sequence, and once at the end. Similarly, the read head was revealed at the beginning and at the end of every output repetition. In all other setups we provide full subtraces (defined for all time steps).

The supervision amount can be tuned by adjusting the $\lambda$ weight from Equation 8. Further, we can also control the fraction of examples which get extra subtrace supervision (the *density* row in Figure I). The performance metric we use is the percentage of runs that do generalize after 100k iterations for the given task and supervision type. By *generalize* we mean that the NTM has perfect accuracy on all testing examples up to $1.5\times$ the size of the max training length, and also perfect accuracy on 90% of the testing examples up to $2\times$ the maximum training length.

We used a feed-forwad controller with $2 \times 50$ units, except for `RepeatCopyTwice`, which uses $2 \times 100$ units. For training we used the Adam optimizer Kingma & Ba (2014), a learning rate of $10^{-3}$ for all tasks except `RepeatFlip3d` and `Flip3rd` which use $5 \cdot 10^{-4}$. The lengths of the training sequences for the first four tasks are from 1 to 5, whereas the generalization of the model was tested with sequences of lengths up to 20. For `Flip3rd` and `RepeatFlip3d`, the training sequence length was up to 16, whereas the testing sequences have maximum length of 32.

## D    SETUP FOR NRAM

Like in the NTM, we use a densely connected two layer feed forward controller for our experiments, and use ReLU as the activation function. We make no modifications to the original architecture, and use noise with the parameter $\eta = 0.3$ as suggested by Neelakantan et al. (2015), and curriculum learning as described by Zaremba & Sutskever (2014). We stop training once we get to a difficulty specified by the task, and increase the difficulty once 0 errors were found on a new testing batch of 10 samples. Each training iteration trains with 50 examples of the currently randomly sampled difficulty. Regardless of whether the model had converged, training is stopped after 5000 samples were used. Such a low number is used to replicate the potential conditions under which such a model might be used. As with the NTM, the Adam optimizer was used. The specific tasks we use are described in Appendix G, and the specific kinds of supervision we give are described in Appendix H. The $\lambda$ we used here was 40. The system was implemented using PyTorch.

## E    NTM Tasks

Every input sequence ends with a special delimiter $x_E$ not occurring elsewhere in the sequence

**Copy** – The input consists of generic elements, $x_1 \ldots x_n x_E$. The desired output is $x_1 \ldots x_n x_E$.

**RepeatCopyTwice** – The input is again a sequence of generic elements, $x_1 \ldots x_n x_E$. The desired output is the input copied twice $x_1 \ldots x_n x_1 \ldots x_n x_E$. Placing the delimiter only at the end of the output ensures that the machine learns to keep track of the number of copies. Otherwise, it could simply learn to cycle through the tape reproducing the given input indefinitely. We kept the number of repetitions fixed in order to increase baseline task performance for the benefit of comparison.

**DyckWords** – The input is a sequence of open and closed parentheses, $x_1 \ldots x_n x_E$. The desired output is a sequence of bits $y_1 \ldots y_n x_E$ such that $y_i = 1$ iff the prefix $x_1 \ldots x_i$ is a balanced string of parentheses (a Dyck word). Both positive and negative examples were given.

**Flip3rd** – The input is a sequence of bits, $x_1 x_2 x_3 \ldots x_n x_E$. The desired output is the same sequence of bits but with the 3rd bit flipped: $x_1 x_2 \bar{x}_3 \ldots x_n x_E$. Such a task with a specific index to be updated (e.g., 3rd) still requires handling data dependence on the contents of the index (unlike say the Copy task).

**RepeatFlip3d** – The input is a sequence of bits, $x_1 x_2 x_3 x_4 x_5 x_5 \ldots x_E$. The desired output is the same sequence of bits but with *every* 3rd bit flipped: $x_1 x_2 \bar{x}_3 x_4 x_5 \bar{x}_6 \ldots x_E$.

## F    NTM Subtraces

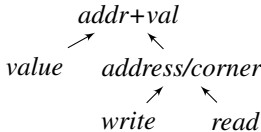

Figure 8: A heirarchy of supervision types (but not quantities) for NTMs.

**value traces**  provide hints for the memory at every timestep as explained in Equation 10.

**read**  – provides a hint for the address of the read head at every timestep.

**write**  – provides a hint for the address of the write head at every timestep.

**address**  – provides hints for the address of both the read and the write head at every timestep.

**addr+val**  – provides value, read and write hints for every timestep.

**corner**  – provides hints for the address of both the read and the write head at every "important" timestep - we decided what important means here depends on which task we are referring to. In general, we consider the first and last timesteps important, and also any timestep where a head should change direction. For example, in RepeatCopyTwice for an example of size $n$ with $e$ repeats, we'd provide the heads at timesteps $0, n, 2n, 3n \ldots, en$.

## G    NRAM TASKS

Below we describe all the tasks we experimented with. We predominantly picked tasks that the NRAM is known to have trouble generalizing on. We did not introduce any new tasks, and more detailed descriptions of these tasks can be found in Kurach et al. (2016).

**Swap** – Provided two numbers, $a$ and $b$ and an array $p$, swap $p[a]$ and $p[b]$. All elements but that in the last memory cell are not zero.

**Increment** – Given an array $p$, return the array with one added to each element. All elements but that in the last cell for the input are not zero. Elements can be zero in the output.

**Permute** – Given two arrays $p$ and $q$ return a new array $s$ such that $s[i] = q[p[i]]$. The arrays $p$ and $q$ are preceded by a pointer, $a$, to array $q$. The output is expected to be $a, s[0] \ldots, s[n], q[0], q[n]$.

**ListK** – Given a linked list in array form, and an index $k$ return the value at node $k$.

**Merge** – given arrays $p$ and $q$, and three pointers $a, b, c$ to array $p$, $q$, and the output sequence (given as zeros initially), place the sorted combination of $p$ and $q$ into the output location.

The following table describes the specific NRAM instantiation used for each task. The *default* sequence (def) is the one described by Kurach et al. (2016). The number of timesteps is usually dependent on the length of the problem instance, $M$ (equivalently the word size or difficulty), and in the case of ListKwas given with respect to the argument $k$. The difficulty (D) was simply the length of the sequence used.

| Task | No. Regs | Module Sequence | Timesteps | Learn Rate | Start D | End D |
|------|----------|-----------------|-----------|------------|---------|-------|
| Swap | 5 | def | 7 | 0.01 | 6 | 10 |
| Increment | 2 | def + R | $M + 2$ | 0.01 | 4 | 10 |
| Permute | 4 | R + def + R + W | $M + 3$ | 0.05 | 7 | 12 |
| ListK | 6 | def | $k + 5$ | 0.05 | 9 | 16 |
| Merge | 8 | def + def + def | $M + 3$ | 0.05 | 13 | 16 |

## H    NRAM SUBTRACES

For each of the tasks listed Appendix G, we hand coded a complete circuit for every module and every timestep we would provide. The following subtraces types describe how we provide hints based on this circuit.

**None** – provides no hints.

**Full** – provides the entire circuit.

**SingleHint** – provides a random hint at a random timestep.

**SingleTimestep** – provides the entire circuit at a random timestep.

**Corners** – provides the entire circuit at the first and last timesteps.

**Registers** – provides hints for the registers at every timestep.

**Modules** – provides hints for the modules at every timestep.

# I NTM RESULTS

| density | | 100 | 100 | 100 | 100 | 50 | 50 | 50 | 50 | 10 | 10 | 10 | 10 | 1 | 1 | 1 | 1 |
|---|---|---|---|---|---|---|---|---|---|---|---|---|---|---|---|---|---|
| λ | | 1 | 0.3 | 0.1 | 0.01 | 1 | 0.3 | 0.1 | 0.01 | 1 | 0.3 | 0.1 | 0.01 | 1 | 0.3 | 0.1 | 0.01 |
| baseline | 52.5 | | | | | | | | | | | | | | | | |
| addr+val | | 100 | 100 | 100 | 70 | 100 | 100 | 100 | 40 | 60 | 80 | 40 | 30 | 30 | 50 | 60 | 10 |
| address | | 100 | 100 | 100 | 50 | 90 | 100 | 90 | 30 | 80 | 90 | 70 | 30 | 50 | 30 | 40 | 50 |
| value | | 100 | 100 | 70 | 40 | 80 | 20 | 40 | 10 | 10 | 20 | 40 | 30 | 60 | 40 | 20 | 10 |
| read | | 90 | 80 | 70 | 50 | 60 | 20 | 50 | 20 | 40 | 40 | 60 | 20 | 70 | 30 | 40 | 10 |
| write | | 60 | 70 | 80 | 60 | 80 | 80 | 40 | 40 | 50 | 70 | 50 | 40 | 50 | 60 | 50 | 40 |
| corner | | 100 | 100 | 100 | 50 | 100 | 90 | 60 | 70 | 70 | 20 | 50 | 30 | 50 | 60 | 20 | 30 |

(a) Copy

| density | | 100 | 100 | 100 | 50 | 50 | 50 | 10 | 10 | 10 | 1 | 1 | 1 |
|---|---|---|---|---|---|---|---|---|---|---|---|---|---|
| λ | | 1 | 0.3 | 0.03 | 1 | 0.3 | 0.03 | 1 | 0.3 | 0.03 | 1 | 0.3 | 0.03 |
| baseline | 15.5 | | | | | | | | | | | | |
| addr+val | | 90 | 100 | 60 | 90 | 80 | 40 | 80 | 20 | 10 | 10 | 0 | 0 |
| address | | 90 | 90 | 90 | 100 | 100 | 40 | 100 | 60 | 0 | 0 | 20 | 30 |
| value | | 80 | 70 | 0 | 50 | 50 | 10 | 30 | 30 | 20 | 10 | 30 | 0 |
| read | | 50 | 30 | 30 | 20 | 60 | 30 | 20 | 60 | 10 | 10 | 10 | 0 |
| write | | 30 | 30 | 20 | 10 | 30 | 40 | 20 | 0 | 10 | 10 | 20 | 20 |
| corner | | 60 | 50 | 40 | 50 | 60 | 10 | 20 | 40 | 20 | 10 | 20 | 0 |

(b) RepeatCopyTwice

| density | | 100 | 100 | 100 | 100 | 50 | 50 | 50 | 50 | 10 | 10 | 10 | 10 | 1 | 1 | 1 | 1 |
|---|---|---|---|---|---|---|---|---|---|---|---|---|---|---|---|---|---|
| λ | | 1 | 0.3 | 0.1 | 0.01 | 1 | 0.3 | 0.1 | 0.01 | 1 | 0.3 | 0.1 | 0.01 | 1 | 0.3 | 0.1 | 0.01 |
| baseline | 45 | | | | | | | | | | | | | | | | |
| address | | 70 | 90 | 60 | 80 | 90 | 90 | 90 | 50 | 80 | 50 | 90 | 80 | 100 | 80 | 70 | 70 |
| read | | 80 | 90 | 70 | 70 | 100 | 100 | 70 | 50 | 50 | 60 | 70 | 70 | 80 | 70 | 50 | 50 |
| corner | | 60 | 100 | 80 | 80 | 80 | 90 | 90 | 90 | 60 | 60 | 100 | 50 | 90 | 80 | 80 | 50 |

(c) DyckWords

| density | | 100 | 100 | 100 | 100 | 50 | 50 | 50 | 50 | 10 | 10 | 10 | 10 | 1 | 1 | 1 | 1 |
|---|---|---|---|---|---|---|---|---|---|---|---|---|---|---|---|---|---|
| λ | | 1 | 0.3 | 0.1 | 0.03 | 1 | 0.3 | 0.1 | 0.03 | 1 | 0.3 | 0.1 | 0.03 | 1 | 0.3 | 0.1 | 0.03 |
| baseline | 45 | | | | | | | | | | | | | | | | |
| addr+val | | 30 | 50 | 50 | 50 | 50 | 50 | 70 | 80 | 60 | 80 | 40 | 40 | 40 | 60 | 40 | 10 |
| address | | 0 | 60 | 40 | 40 | 20 | 70 | 80 | 90 | 90 | 70 | 60 | 50 | 50 | 50 | 60 | 60 |
| value | | 60 | 50 | 40 | 60 | 80 | 70 | 40 | 10 | 50 | 10 | 40 | 70 | 50 | 30 | 40 | 60 |
| read | | 40 | 60 | 70 | 40 | 30 | 80 | 90 | 90 | 100 | 70 | 80 | 50 | 30 | 50 | 60 | 30 |
| write | | 0 | 30 | 50 | 20 | 30 | 40 | 60 | 20 | 40 | 40 | 40 | 40 | 20 | 50 | 70 | 50 |
| corner | | 50 | 70 | 80 | 80 | 40 | 40 | 90 | 70 | 80 | 70 | 60 | 40 | 50 | 60 | 60 | 40 |

(d) Flip3rd

| density | | 100 | 100 | 100 | 100 | 50 | 50 | 50 | 50 | 10 | 10 | 10 | 10 | 1 | 1 | 1 | 1 |
|---|---|---|---|---|---|---|---|---|---|---|---|---|---|---|---|---|---|
| λ | | 1 | 0.5 | 0.1 | 0.03 | 1 | 0.5 | 0.1 | 0.03 | 1 | 0.5 | 0.1 | 0.03 | 1 | 0.5 | 0.1 | 0.03 |
| baseline | 5 | | | | | | | | | | | | | | | | |
| addr+val | | 30 | 20 | 20 | 40 | 30 | 30 | 10 | 10 | 40 | 10 | 0 | 20 | 10 | 10 | 0 | 10 |
| address | | 20 | 50 | 30 | 30 | 30 | 40 | 20 | 40 | 20 | 40 | 20 | 0 | 0 | 0 | 20 | 0 |
| value | | 0 | 0 | 20 | 20 | 0 | 0 | 0 | 0 | 10 | 10 | 10 | 0 | 0 | 0 | 0 | 0 |
| read | | 30 | 10 | 40 | 20 | 10 | 30 | 20 | 40 | 30 | 10 | 0 | 10 | 20 | 0 | 10 | 20 |
| write | | 0 | 0 | 0 | 10 | 0 | 0 | 0 | 0 | 10 | 10 | 0 | 0 | 20 | 20 | 30 | 0 |
| corner | | 40 | 40 | 60 | 20 | 50 | 30 | 10 | 30 | 10 | 10 | 0 | 0 | 0 | 10 | 0 | 10 |

(e) RepeatFlip3d

Figure 9: Baselines have generalization on over 40 different initial weights. Other tests use 10.

**Which Details to Reveal for NTM?**    The first dimension listed in the rows of the tables of Figure I controls the execution details revealed in a Subtrace. We use subtraces showing either the *addresses* without the tape *values*, only the *read heads* or the *write heads*, or even weaker supervision in a few *corner* cases. In tasks Copy Figure 9a), RepeatCopyTwice (Figure 9b) and DyckWords (Figure 9c), it is frequently the case that when the NTM generalizes without supervision, it converges to an algorithm which we are able to interpret. For them, we designed the *addr+val* traces to match this algorithm, and saw increases in generalization frequency of at least 45%. It can be concluded that when additionally provided supervision reflects the interpretable "natural" behavior of the NTM, the learning becomes significantly more robust to changes in initial weights. Additionally, for tasks Flip3rd (Figure 9d) and RepeatFlip3d (Figure 9e), both the baseline and other supervision types are outperformed by training with *read* supervision. It is also notable that *corner* supervision in RepeatFlip3d achieves highest improvement over the baseline, 60% over 5%. In essence, this means that providing only a small part of the trace can diminish the occurrence of local minima in the loss function.

**How Often to Reveal for NTM?**    The second dimension controls the proportion of examples that receive extra subtrace supervision (the *density* columns in Figure I). For Flip3rd, RepeatCopyTwice and DyckWords we observed that having only a small number of examples with extra supervision leads to models which are more robust to initial weight changes than the baseline, although not necessarily always as robust as providing supervision all the time.

A couple of interesting cases stand out. For Flip3rd with 10% *corner* subtraces and $\lambda = 1$, we find a surprisingly high rate of generalization. Providing *address* traces 10% of the time when training RepeatCopyTwice leads to better performance all the time. For RepeatFlip3d, *write* traces at 1% frequency and $\lambda = 0.1$ generalize 30% of the time vs. 5% for baseline.

While the type of trace which works best varies per task, for each task there exists a trace which can be provided only 1% of the time and still greatly improve the performance over the baseline. This suggests that a small amount of extra supervision can improve performance significantly, but the kind of supervision may differ. It is an interesting research question to find out how the task at hand relates to the optimal kind of supervision.

## J  NRAM RESULTS

| Subtrace Type \Task | Permute | Swap | Increment | ListK | Merge | PermuteNoise |
|---|---|---|---|---|---|---|
| None | 36 | 44 | 58 | 41 | 13 | 12 |
| SingleHint | 24 | 38 | 28 | 22 | 12 | 14 |
| Corners | 29 | 23 | 36 | 22 | 9 | 17 |
| SingleTimestep | 21 | 52 | 29 | 28 | 12 | 13 |
| Registers | 48 | 58 | 73 | 54 | - | - |
| Modules | 48 | 58 | 107 | 54 | - | - |
| Full | 26 | 33 | 32 | 44 | 21 | 14 |
| NoisyFull | - | - | - | 36 | - | - |

Figure 10: The number of runs which completed for each task and subtrace type. The Data in the graphs below is taken by averaging the results of these runs.

| Subtrace Type \Task | Permute | Swap | Increment | ListK |
|---|---|---|---|---|
| None | 6290.29 | 5505.22 | 3500.13 | 5880.11 |
| SingleHint | 5565.22 | 3700.64 | 4318.20 | 6574.59 |
| Corners | 4468.85 | 6195.75 | 3199.86 | 6601.16 |
| SingleTimestep | 6259.05 | 2662.35 | 4042.18 | 5076.17 |
| Registers | 6618.12 | 5774.61 | 3839.18 | 6185.54 |
| Modules | 6523.16 | 5781.99 | 2335.99 | 6183.74 |
| Full | 4919.33 | 4110.14 | 3758.99 | 3216.01 |

Figure 11: The average time (in seconds) to finish training for each task and subtrace type. For most tasks it is clear that Full traces while introducing extra computations to individual timesteps, reduce the amount of time to finish training over not using supervision.

| Subtrace Type \Task | ListK | Swap | Permute | Increment | Merge | PermuteNoise |
|---|---|---|---|---|---|---|
| None | 95.08 | 91.52 | 99.97 | 99.91 | 99.96 | 99.99 |
| SingleHint | 93.61 | 2.41 | 57.55 | 14.86 | 100.0 | 56.90 |
| Corners | 94.47 | 99.09 | 16.40 | 2.14 | 100.0 | 20.79 |
| SingleTimestep | 36.91 | 1.75 | 47.79 | 13.77 | 100.0 | 33.60 |
| Full | 12.77 | 11.01 | 7.83 | 9.89 | 78.44 | 23.57 |
| Registers | 93.22 | 93.44 | 99.97 | 90.36 | - | - |
| Modules | 93.70 | 95.57 | 86.48 | 40.87 | - | - |

Figure 12: The average number of errors on the test set for each task and subtrace type once trained.

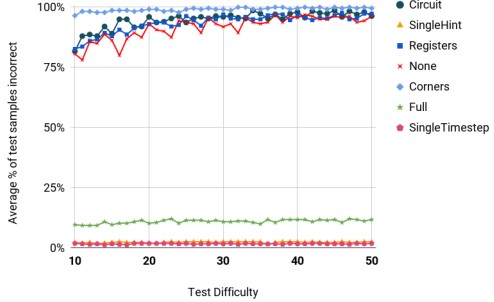

Figure 13: Comparing average generalization to sequence length for Swap

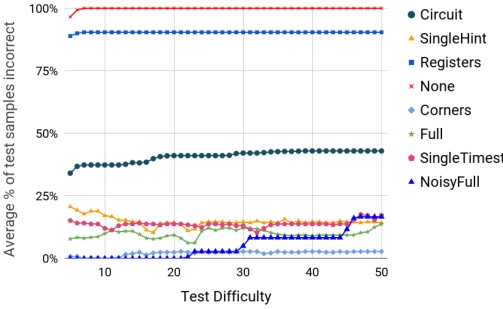

Figure 14: Comparing average generalization to sequence length for Increment

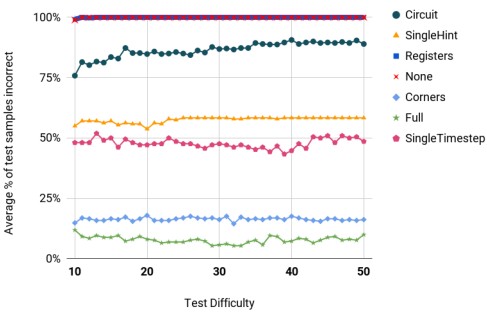

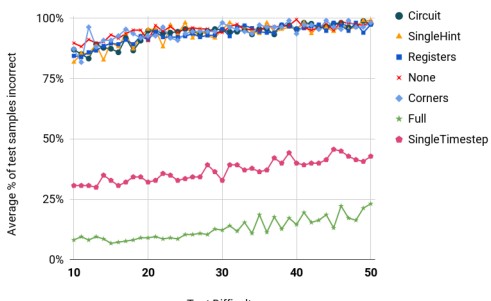

Figure 15: Comparing average generalization to sequence length for `Permute`

Figure 16: Comparing average generalization to sequence length for `ListK`

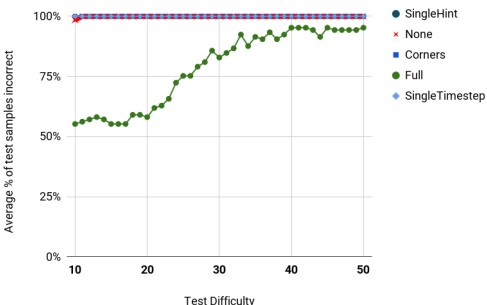

Figure 17: Comparing average generalization to sequence length for `Merge`

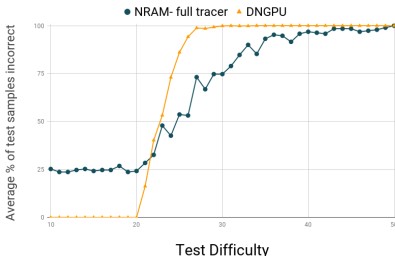

Figure 18: Comparing average generalization of the DNGPU Freivalds & Liepins (2017) with that of the NRAM using the full tracer for `Merge`. For this experiment, a maximum of 10000 samples were used for the DNGPU and 5000 for the NRAM. The DNGPU was run out of the box from the code supplied by the authors. 20 runs were averaged for the DNGPU and 38 runs for the NRAM. One can deduce that while neither is able to generalize this task perfectly, the simpler and easier to understand architecture, NRAM, does generalize better with fewer examples when those examples come with richer supervision.

## K  PROGRAMMING NRAMS

The NRAM is parametrized by one or more straight-line partial programs, i.e., programs with no branching and no loops, chosen by register states. The machine runs in a loop, repeatedly selecting the program for that register state then executing it. The programs are expressend in a simple single-assignment imperative language. Each program statement $i$ invokes one of the modules of the architecture and assigns the result of the invocation to a local variable $x_i$. That variable cannot be changed later. The final program statement is a parallel-asignment that modifies the machine registers $r_1 \ldots r_k$. The values that appear in assignments/invocations can be: variables in scope, machine registers, or holes ?. These values are not used directly during execution: the actual values needs to be supplied by the NRAM controller. The values are only used as hints for the controller during training, with the whole ? denoting no hint. We can describe the language in an EBNF-style grammar:

$$F_n ::= \text{modules of arity } n \qquad\qquad S_i ::= x_i \leftarrow F_n(\underbrace{V_i, \ldots, V_i}_{n}) \qquad (17)$$

$$V_i ::= r_1 \mid \cdots \mid r_k \mid ? \mid x_1 \mid \ldots \mid x_{i-1} \qquad R_i ::= r_{j_1}, \ldots, r_{j_n} \leftarrow \underbrace{V_i, \ldots V_i}_{n} \qquad (18)$$

$$P_1 ::= S_1 \quad P_i ::= P_{i-1}; S_i \qquad\qquad P ::= P_1; R_1 \mid P_2; R_2 \mid \ldots \qquad (19)$$

An example program for the **Increment** task would be the following:

$x_1 \leftarrow 1;$
$x_2 \leftarrow \text{READ}(r_1);$
$x_3 \leftarrow \text{ADD}(x_2, x_1);$
$x_4 \leftarrow \text{WRITE}(r_1, x_3);$
$x_5 \leftarrow \text{ADD}(r_1, x_1);$
$r_1 \leftarrow x_5$

Here, the controller is encouraged to read the memory at the location stored in the first register $r_1$, add one to it, then store it back, and then increment the the first register.

An alternative to the trace-based approach is to make the controller produce values only for the holes, and use directly the specified variable/register arguments. This way, only the unspecified parts of the program are learned. This is, for example, the approach taken by $\partial$Forth Bošnjak et al. (2017). There, programs are expressed in a suitably adapted variant of the Forth programming language, which is as expressive as the language discussed above, but less syntactically constrained.

The drawback of this alternative is that whenever an argument other than a whole is specified, one must also specify the time steps to which it applies in *all possible executions* and not just the training ones. That is why, typically, these values are specified either for all or for none of the time steps.

In the following examples, we will describe the register states using "0", "!0" and "-" meaning respectively that a register has 0, that it contains anything but zero, or that it can contain anything.

## L  NRAM PERMUTATION PROGRAM

For any register pattern.

$x_1 \leftarrow READ(r_0);$
$x_2 \leftarrow WRITE(0, x_1);$
$x_3 \leftarrow READ(r_1);$
$x_4 \leftarrow ADD(x_3, x_1);$
$x_5 \leftarrow READ(x_4);$
$x_6 \leftarrow WRITE(r_1, x_5);$
$x_7 \leftarrow INC(r_1);$
$x_8 \leftarrow DEC(x_1);$
$x_9 \leftarrow LT(x_7, x_8);$
$r_0 \leftarrow 0;$
$r_1 \leftarrow x_7;$
$r_2 \leftarrow x_9;$
$r_3 \leftarrow 0;$

## M   NRAM LISTK PROGRAM

When the registers are [0, !0, !0, -, -]:
$x_1 \leftarrow READ(r_0);$
$x_2 \leftarrow ADD(x_1, 2);$
$x_3 \leftarrow WRITE(0, x_1);$
$r_0 \leftarrow 1;$
$r_1 \leftarrow 1;$
$r_2 \leftarrow 1;$
$r_3 \leftarrow x_2;$

When the registers are [!0, !0, !0, -, -]:
$x_1 \leftarrow READ(r_1);$
$x_2 \leftarrow ADD(x_1, 2);$
$x_3 \leftarrow WRITE(r_1, x_1);$
$r_0 \leftarrow 1;$
$r_1 \leftarrow 0;$
$r_2 \leftarrow 1;$
$r_3 \leftarrow r_3;$
$r_4 \leftarrow x_2;$

When the registers are [!0, 0, !0, -, -]:
$x_1 \leftarrow READ(r_3);$
$x_2 \leftarrow WRITE(r_3, x_1);$
$r_0 \leftarrow 1;$
$r_1 \leftarrow 0;$
$r_2 \leftarrow 0;$
$r_3 \leftarrow x_1;$
$r_4 \leftarrow 4;$

When the registers are [!0, 0, 0, -, -]:
$x_1 \leftarrow READ(r_4);$
$x_2 \leftarrow WRITE(r_4, r_3);$
$r_0 \leftarrow 1;$
$r_1 \leftarrow 1;$
$r_2 \leftarrow 0;$
$r_3 \leftarrow x_1;$

When the registers are [0, !0, 0, -, -]:
$x_1 \leftarrow READ(r_2);$
$x_2 \leftarrow ADD(x_1, 2);$
$x_3 \leftarrow WRITE(x_2);$
$r_0 \leftarrow 0;$
$r_1 \leftarrow 1;$
$r_2 \leftarrow 0;$

## N   NRAM LISTK PROGRAM

Timestep 0:
$x_1 \leftarrow READ(r_0);$
$x_2 \leftarrow INC(x_1);$
$x_3 \leftarrow 0;$
$x_4 \leftarrow WRITE(x_3, x_1);$
$r_0 \leftarrow r_1;$
$r_1 \leftarrow r_1;$
$r_2 \leftarrow r_2;$
$r_3 \leftarrow x_2;$
$r_4 \leftarrow r_4;$

$r_5 \leftarrow r_5;$

Timestep 1:

$x_1 \leftarrow READ(r_1);$
$x_2 \leftarrow WRITE(r_1, x_1);$
$x_3 \leftarrow 0;$
$r_0 \leftarrow r_0;$
$r_1 \leftarrow x_1;$
$r_2 \leftarrow r_2;$
$r_3 \leftarrow r_3;$
$r_4 \leftarrow x_3;$
$r_5 \leftarrow r_5;$

Timestep 2:

$x_1 \leftarrow READ(r_2);$
$x_2 \leftarrow WRITE(r_2, x_1);$
$x_3 \leftarrow 0;$
$r_0 \leftarrow r_0;$
$r_1 \leftarrow r_1;$
$r_2 \leftarrow x_1;$
$r_3 \leftarrow r_3;$
$r_4 \leftarrow x_3;$
$r_5 \leftarrow x_3;$

Timestep 3 to 3 + k - 1:

$x_1 \leftarrow READ(r_0);$
$x_2 \leftarrow INC(x_1);$
$x_3 \leftarrow 0;$
$x_4 \leftarrow DEC(x_1);$
$x_5 \leftarrow WRITE(r_0, x_1);$
$r_0 \leftarrow x_1;$
$r_1 \leftarrow x_4;$
$r_2 \leftarrow r_2;$
$r_3 \leftarrow x_2;$
$r_4 \leftarrow x_3;$
$r_5 \leftarrow x_3;$

Timestep 3 + k:

$x_1 \leftarrow READ(r_3);$
$x_2 \leftarrow WRITE(r_2, x_1);$
$x_3 \leftarrow 0;$
$x_4 \leftarrow 1;$
$r_0 \leftarrow r_0;$
$r_1 \leftarrow r_1;$
$r_2 \leftarrow r_2;$
$r_3 \leftarrow x_1;$
$r_4 \leftarrow x_4;$
$r_5 \leftarrow x_3;$

Rest:

$x_1 \leftarrow WRITE(r_2, r_3);$
$x_2 \leftarrow 1;$
$x_3 \leftarrow 0;$
$r_0 \leftarrow r_0;$
$r_1 \leftarrow r_1;$
$r_2 \leftarrow r_2;$
$r_3 \leftarrow r_3;$
$r_4 \leftarrow x_2;$
$r_5 \leftarrow x_3;$

