# OpenReview forum: "Training Neural Machines with Partial Traces"
_ICLR.cc/2018/Conference — Reject_

### Official Review · AnonReviewer1 · 2017-11-26
**Interesting synthesis for neural programming machines lacking strong baselines.**

**Rating:** 4
**Confidence:** 5

**Review:**

The authors introduce the general concept of a differential neural computational machine, dNCM. It can apply to any fully differentiable neural programming machine, such as the Neural Turing Machine (NTM) or NRAM or the Neural GPU, but not to non-fully-differentiable architecture such as NPI. The author show how partial traces can be used to improve training of any dNCM with results on instantiations of dNCM for NTM and NRAM.

On the positive side, the paper is well-written (though too many results require looking into the Appendix) and dNCM is elegant. Also, while it's hard to call the idea of using partial traces original, it's not been studied in this extent and setting before. On the negative side, the authors have chosen weak baselines and too few and easy tasks to be sure if their results will actually hold in general. For example, for NTM the authors consider only 5 tasks, such as Copy, RepeatCopyTwice, Flip3rd and so on (Appendix E) and define  "a model to generalize if relative to the training size limit n, it achieves perfect accuracy on all of tests of size ≤ 1.5n and perfect accuracy on 90% of the tests of size ≤ 2n". While the use of subtraces here shows improvements, it is not convincing since other architectures, e.g., the Improved Neural GPU (https://arxiv.org/abs/1702.08727), would achieve 100% on this score without any need for subtraces or hints. The tasks for the NRAM are more demanding, but the results are also more mixed. For one, it is worrysome that the baseline has >90% error on each task (Appendix J, Figure 12) and that Merge even with full traces has still almost 80% errors. Neural programmers are notoriously hard to tune, so it is hard to be sure if this difference could be eliminated with more tuning effort. In conclusion, while we find this paper valuable, to be good enough for acceptance it should be improved with more experimentation, adding baselines like the (Improved) Neural GPU and more tasks and runs.

---

> ### Author Response · Authors · 2018-01-05
> **Additional Experiments**
>
>
>
> → More tasks and runs are required.
>
> We performed additional experiments with noise in the supervision and also with other architectures (NGPU) showing benefits. We updated the paper with these results.

---

### Official Review · AnonReviewer3 · 2017-11-27
**Useful abstraction?**

**Rating:** 5
**Confidence:** 4

**Review:**

Summary

This paper presents differentiable Neural Computational Machines (∂NCM), an abstraction of existing neural abstract machines such as Neural Turing Machines (NTMs) and Neural Random Access Machines (NRAMs). Using this abstraction, the paper proposes loss terms for incorporating supervision on execution traces. Adding supervision on execution traces in ∂NCM improves performance over NTM and NRAM which are trained end-to-end from input/output examples only. The observation that adding additional forms of supervision through execution traces improves generalization may be unsurprising, but from what I understand the main contribution of this paper lies in the abstraction of existing neural abstract machines to ∂NCM. However, this abstraction does not seem to be particularly useful for defining additional losses based on trace information. Despite the generic subtrace loss (Eq 8), there is no shared interface between ∂NCM versions of NTM and NRAM that would allow one to reuse the same subtrace loss in both cases. The different subtrace losses used for NTM and NRAM (Eq 9-11) require detailed knowledge of the underlying components of NTM and NRAM (write vector, tape, register etc.), which questions the value of ∂NCM as an abstraction.

Weaknesses

As explained in the summary, it is not clear to me why the abstraction to NCM is useful if one still needs to define specific subtrace losses for different neural abstract machines.
The approach seems to be very susceptible to the weight of the subtrace loss λ, at least when training NTMs. In my understanding each of the trace supervision information (hints, e.g. the ones listed in Appendix F) provides a sensible inductive bias we would the NTM to incorporate. Are there instances where these biases are noisy, and if not, could we incorporate all of them at the same time despite the susceptibility w.r.t λ?
NTMs and other recent neural abstract machines are often tested on rather toyish algorithmic tasks. I have the impression providing extra supervision in form of execution traces makes these tasks even more toyish. For instance, when providing input-output examples as well as the auxiliary loss in Eq6, what exactly is left to learn? What I like about Neural-Programmer Interpreters and Neural Programmer [1] is that they are tested on less toyish tasks (a computer vision and a question answering task respectively), and I believe the presented method would be more convincing for a more realistic downstream task where hints are noisy (as mentioned on page 5).

Minor Comments

p1: Why is Grefenstette et al. (2015) an extension of NTMs or NRAMs? While they took inspiration from NTMs, their Neural Stack has not much resemblance with this architecture.
p2: What is B exactly? It would be good to give a concrete example at this point. I have the feeling it might even be better to explain NCMs in terms of the communication between κ, π and M first, so starting with what I, O, C, B, Q are before explaining what κ and π are (this is done well for NTM as ∂NCM in the table on page 4). In addition, I think it might be better to explain the Controller before the Processor. Furthermore, Figure 2a should be referenced in the text here.
p4 Eq3: There are two things confusing in these equations. First, w is used as the write vector here, whereas on page 3 this is a weight of the neural network. Secondly, π and κ are defined on page 2 as having an element from W as first argument, which are suddenly omitted on page 4.
p4: The table for NRAM as ∂NCM needs a bit more explanation. Where does {1}=I come from? This is not obvious from Appendix B either.
p3 Fig2/p4 Eq4: Related to the concern regarding the usefulness of the ∂NCM abstraction: While I see how NTMs fit into the NCM abstraction, this is not obvious at all for NRAMs, particularly since in Fig 2c modules are introduced that do not follow the color scheme of κ and π in Fig 2a (ct, at, bt and the registers).
p5: There is related work for incorporating trace supervision into a neural abstract machine that is otherwise trained end-to-end from input-output examples [2].
p5: "loss on example of difficulties" -> "loss on examples of the same difficulty"
p5: Do you have an example for a task and hints from a noisy source?
Citation style: sometimes citation should be in brackets, for example "(Graves et al. 2016)" instead of "Graves et al. (2016)" in the first paragraph of the introduction.

[1] Neelakantan et al. Neural programmer: Inducing latent programs with gradient descent. ICLR. 2015.
[2] Bosnjak et al. Programming with a Differentiable Forth Interpreter. ICML. 2017.

---

> ### Author Response · Authors · 2018-01-05
> **Using the Abstraction**
>
>
>
> → What is left to learn if I/O examples and auxiliary loss (Eq 6) are already provided?
>
> Even for full supervision, the actual values in the registers are not provided, and the computation halting time step is not provided.
>
> → The approach would be more impressive if it was trained on less toyish tasks.
>
> We agree that more complex tasks would be even better. However, even the current tasks (also studied by NRAM and dForth) are challenging and show the benefits of our supervision method.
>
> → It is unclear how NRAMs fit into ∂NCMs.
>
> The NRAM factorizes into a neural controller that communicates in rounds with a non-neural circuitry as pictured in Figure 2.  This is the factorization needed by a ∂NCM, and the particular instantiation is given in equation (4).   We will update the text to elaborate a little more here.

---

### Official Review · AnonReviewer2 · 2017-11-27
**Insights not clear enough**

**Rating:** 4
**Confidence:** 3

**Review:**

Much of the work on neural computation has focused on learning from input/output samples.  This paper is a study of the effect of adding additional supervision to this process through the use of loss terms which encourage the interpretable parts of the architecture to follow certain expected patterns.

The paper focuses on two topics:
1.  Developing a general formalism for neural computers which includes both the Neural Turing Machine (NTM) and the Neural Random Access Machine (NRAM), as well as a model for providing partial supervision to this general architecture.

2.  An experimental study of providing various types of additional supervision to both the NTM and the NRAM architecture.

I found quite compelling the idea of exploring the use of additional supervision in neural architectures since oftentimes a user will know more about the problem at hand than just input-output examples.  However, the paper is focused on very low-level forms of additional supervision, which requires the user to deeply understand the neural architecture as well as the way in which a given algorithm might be implemented on this architecture.  So practically speaking I don't think it's reasonable to assume that users would actually provide additional supervision in this form.

This would have been fine, if the experimental results provided some insights into how to extend and/or improve existing architectures.  Unfortunately,  the results were simply a very straight-forward presentation of a lot of numbers, and so I was unable to draw any useful insights.  I would have liked the paper to have been more clear about the insights provided by each of the tables/graphs.  In general we can see that providing additional supervision improves the results, but this is not so surprising.

Finally, much of the body of the paper is focused on topic (1) from above, but I did not feel as though this part of the paper was well motivated, and it was not made clear what insights arose from this generalization.  I would have liked the paper to make clear up front the insights created by the generalization, along with an intuitive explanation.  Instead much of the paper is dedicated to introduction of extensive notation, with little clear benefit.  The notation did help make clear the later discussion of the experiments, but it was not clear to me that it was required in order to explain the experimental results.

So in summary I think the general premise of the paper is interesting, but in it's current state I feel like the paper does not have enough sufficiently clear insights for acceptance.

---

> ### Author Response · Authors · 2018-01-05
> **Our Insights**
>
>
> → It is unclear what the experimental insights of the paper are.
>
> We believe a key result is that providing relatively simple hints on the interpretable part of the architecture leads to significantly improved results. This is particularly important with neural programmers which are notoriously difficult to tune (as pointed by Reviewer 3). To a degree, ability to provide additional supervision eliminates some of the complex and time consuming tuning process and makes the architecture more robust. We also have demonstrated that the supervision with NRAM can lead to better results than state-of-the-art architectures such as nGPU.

---

### Author Response · Authors · 2018-01-05
**Main Questions**



→ Supervision requires the user to know the architecture and how the algorithm would be implemented on that architecture. The generalization is not well motivated and its insights are unclear.  Why is the ∂NCM abstraction useful if supervision requires detailed knowledge of the architecture?

To provide extra supervision, the user needs to be aware of the general architecture, but they need only know in detail what the interpretable portion is. We demonstrate that this is reasonable trade-off as a little bit of extra knowledge can enable substantially better results. Architectures are often heavily tuned to particular classes of tasks, which already requires deep knowledge of the machinery.

We see ∂NCM as a mechanism for explaining our form of supervision and to which architectures it applies and how supervision works. It is not meant as a general purpose model for specifying only supervision at the ∂NCM level, without being aware of the underlying architecture. Thus, we believe the ∂NCM abstraction is useful for understanding purposes.

→ How does your work compare to Differential Forth (dForth) (ICML’17) ? This work already provides a form of supervision.

The main conceptual difference between our work and dForth is the kind of supervision provided. In dForth, the teacher provides a static sketch program and leaves only the holes in the sketch to be learned. In our work, there is no sketch with holes: the teacher provides hints (soft constraints) for some time steps as a sequence of instructions to be executed. The controller then has to learn to issue the correct sequence at every time step.

Both are interesting and different ways of providing supervision and both have been studied in traditional non-neural programming by example synthesis, referred to as sketch vs. trace-based supervision.

→ Why your NRAM baseline results do not match the NRAM results in the original paper?

Generally, our metric of success is harder to satisfy. In more detail:

(i) Our metric of success tolerates less errors than the metric in the NRAM paper, meaning they can report success where we report failure. We believe our metric is a more natural one (see below).

The  NRAM paper says: “The final error for a test case is computed as c / m, where c is the number of correctly written cells, and m represents the total number of cells that should be modified”. While not explained in that paper, a conservative interpretation is that cells that should be modified are specified by the task. Merge, for example, could specify the last half of the input string as to be modified.  Errors in spots that shouldn’t be modified are not counted.

On the other hand, in our paper, we consider correctness of the entire string at once. We do not tolerate a single wrong entry.

(ii) We show graphs of the error up to examples of size 50 while their graphs are for samples of size 30.  Further, points on the horizontal axis in their graph represent not just the generalization on examples of that difficulty, but the average generalization of examples of up to that difficulty, computed by averaging an example of uniformly random difficulty of at most that point.

(iii) our graphs represent the average error over multiple trainings, while it is not specified if theirs are the result of multiple trainings or just one.

→ Why are subtraces useful if the improved Neural GPU (NGPU) can solve some of this tasks without supervision?

Good point. Based on the reviewer’s suggestion we investigated this question, and updated the Appendix to include the results for the harder Merge task (see Figure 18). It turns out that while the NGPU can provide better baseline results than NRAM without supervision, the NRAM with supervision can have substantially better results than NGPU. Such results are possible only because the NRAM is more interpretable than the NGPU, allowing extra supervision and thus, better results.

For the simpler tasks, the NGPU sometimes generalizes perfectly (flip-3rd, repeat-flip-third, increment, swap), but often it generalizes worse (permute, list-k, dyck-words) than our supervision method.

→ How do you provide supervision with NRAM exactly?

We added an Appendix describing the NRAM programming language used to provide supervision and also provided examples of supervision at different time steps in that language to be executed at time steps determined from a pre-condition on the state of the memory.

→ The approach would be more convincing if it considered noisy hints.

Based on the reviewers’ suggestion we now updated the paper with an experiment which shows that the presence of noisy and corrupted hints still significantly outperforms unsupervised training in the case of the increment task.   The line “NoisyFull” in Figure 14 demonstrates that even with corrupted hints, supervision substantially helps training.  This experiment is explained in more detail in “Robustness to Noise” in Section 5.

---

### Author Response · Authors · 2018-01-05
**Overview of changes**

We thank the reviewers for their comments. We addressed many of the major comments by performing additional experiments, and updated the Appendix in the paper to reflect that:

- A comparison to the state-of-the-art Neural GPU (NGPU) showing NRAM with trace supervision can produce better results than NGPU.

- Additional experiments with noise in the supervision, showing that even with noise, we can produce better results than the no supervision NRAM baseline.

- A description of the supervision language for NRAM together with supervision examples.

---

### Decision · Program_Chairs · 2018-01-29
**ICLR 2018 Conference Acceptance Decision**

**Decision:**

Reject

**Comment:**

While the reviewers considered the basic idea of adding supervision intermediate to differentiable programming style architectures to be interesting and worthy of effort, they were unsure if
1: the proposed abstractions for discussing ntm and nram are well motivated/more generally applicable
2: the methods used in this work to give intermediate supervision are more generally applicable